# Exploring how lifestyle weight management programmes for children are commissioned and evaluated in England: a mixed methodology study

Ruth Mears [1,2] Russ Jago [2] Deborah Sharp,[1] Anamica Patel,[2]
Ruth Kipping,[3,4] Julian P H Shield[5,6]

For numbered affiliations see end of article.

**Correspondence to**
Dr Ruth Mears;
rm14101@bristol.ac.uk

## ABSTRACT

**Objective** To assess how lifestyle weight management programmes for children aged 4–16 years in England are commissioned and evaluated at the local level.

**Design** This was a mixed-methods study comprising an online survey and semistructured telephone interviews.

**Setting** An online survey was sent to all local authorities (LAs) in England regarding lifestyle weight management services commissioned for children aged 4–16 years. Online survey data were collected between February and May 2016 and based on services commissioned between April 2014 and March 2015. Semistructured telephone interviews with LA staff across England were conducted between April and June 2016.

**Participants** Commissioners or service providers working within the public health department of LAs.

**Main outcome measures** The online survey collected information on the evidence base, costs, reach, service usage and evaluation of child lifestyle weight management services. The telephone interviews explored the nature of child weight management contracts commissioned by LAs, the type of outcome data collected and whether these data were shared with other LAs or organisations, the challenges faced by these services, and the perceived 'markers of success' for a programme.

**Results** The online survey showed that none of the participating LAs was aware of any peer-reviewed evidence supporting the effectiveness of their specific commissioned service. Despite this, the telephone interviews revealed that there was no national formal sharing of data to enable oversight of the effectiveness of commissioned services across LAs in England to help inform future commissioning decisions. Challenges with long-term data collection, service engagement, funding and the pressure to reduce the prevalence of obesity were frequently mentioned.

**Conclusions** Robust, independent, cost-effectiveness analyses of obesity strategies are needed to determine the appropriate allocation of funding to lifestyle weight management treatment services, population-level preventative approaches or development of whole system approaches by an LA.

## INTRODUCTION

In the past four decades, there has been a 10-fold increase in the number of obese

### Strengths and limitations of this study

▶ There has been no previous independent, peer-reviewed research study assessing how lifestyle weight management programmes in childhood are being commissioned and evaluated across local authorities (LAs) in England.

▶ The response rate for the online survey was lower than desired; however, there was good geographical representation across England.

▶ The current study focused on LAs in England, so generalisation of results to the rest of the UK and wider is unclear.

▶ The change in weight status and cost data provided by LAs precluded meaningful statistical analyses, so it is impossible to comment on the cost-effectiveness of, or between, commissioned services.

▶ There were no freedom of information requests submitted to LAs who did not complete the online survey, and it is possible further data could have been obtained through this route.

children and adolescents worldwide.[1 2] In the UK, 31.1% of children and adolescents were classified as overweight or obese in 2016.[3] These children and adolescents are more likely to become overweight or obese adults and suffer health-related consequences.[4] This presents a major public health challenge.[5] In the UK, weight management strategies are classified into tier 1 (those that focus on preventing obesity), tier 2 (lifestyle weight management services), tier 3 (specialist obesity services) and tier 4 (pharmacological or surgical treatments for obesity) services.[6] Tier 1 and 2 services are commissioned by public health departments working within local authorities (LAs). Clinical commissioning groups (CCGs) are responsible for commissioning tier 3 services since 2014 and tier 4 services since 2017.[7] CCGs are responsible for the planning and commissioning of healthcare services for their local area and

are assured by NHS England.[8] In 2013, Public Health England (PHE) was formed as a separate entity to NHS England as public healthcare transitioned from the National Health Service (NHS) to LAs under the Health and Social Care Act 2012.[9]

This paper focuses on tier 2 weight management services commissioned by LAs across England for school-aged children (aged 4–16 years). There are 152 LAs in England,[10] and each LA may choose to commission services from a different tier 2 service provider. Although there is guidance from the National Institute for Health and Care Excellence (NICE) and PHE regarding what these services should comprise,[6 11] the specific weight management programmes have rarely been independently evaluated and published. Furthermore, there are very few UK-based, randomised trials in the peer-reviewed literature demonstrating a clinically significant reduction in body mass index (BMI) z-score (defined as minimum BMI Standard Deviation Score reduction of ≥0.25)[12 13] through lifestyle weight management programmes alone for school-aged children.[14–16] Even the evidence reviews supporting the NICE public health guidance (PH47) only reported a postintervention pooled reduction in BMI z-score of −0.17 (95% CI −0.3 to −0.04, p=0.01), which was attenuated when long-term data (≥6 months) were used (standardised mean difference=−0.07; 95% CI −0.15 to 0.02, p=0.12).[17]

LAs usually monitor their tier 2 weight management services through 'Performance Management' meetings, although they may also conduct service evaluations. NICE recommends that monitoring focuses on sustaining long-term changes,[6] despite their evidence reviews showing little efficacy for these interventions in the long term.[17] Given the poor evidence base for tier 2 weight management services, it is important to understand more about the nature of the contracts commissioned by LAs, the monitoring of outcomes and the challenges facing these services. In addition, given the current financial climate in public health, with spending estimated by the King's Fund to be 8% lower 4 years after public health moved from the NHS to LAs,[18] it is important to explore whether these services are a good use of limited resources.

This mixed-methods study uses quantitative methods (an online survey) to determine the evidence base underpinning the local service provided, costs, reach, service usage and evaluation of tier 2 weight management programmes commissioned by LAs across England for children aged 4–16 years. Qualitative methods (semistructured telephone interviews) explore the nature of childhood tier 2 weight management contracts commissioned by LAs, the type of outcome data collected and whether these data are shared, the challenges faced by these services, and the perceived 'markers of success' for a programme. Finally, the data collected from both the online survey and telephone interviews examine whether lifestyle weight management programmes are a good use of limited resources.

## METHODS

### Participants and recruitment

A list of all 152 LAs in England was derived from 2014/2015 National Child Measurement Programme data sets.[19] The Director for Public Health for each LA was contacted by email and asked to identify the relevant person within their LA responsible for the commissioning of childhood tier 2 weight management services. An email was sent to this person asking if they would be willing to participate in an online survey exploring tier 2 weight management services for school-aged children commissioned between April 2014 and March 2015. If no response to the email was received, a further email was sent.

The final page of the online survey provided information about the second phase of the study (telephone interviews) and invited those interested in taking part to leave their contact details. In addition, some of those LAs who declined to participate in the online survey were also invited by email to take part in the telephone interviews.

### Patient and public involvement

Patients and the public were not involved in the design of this study or interpretation of results.

### Design of online survey and telephone interview guide

The online survey (online supplementary file 1) and interview guide (online supplementary file 2) were developed by RM, RJ, DS, JPHS and RK. Development of the survey and interview guide was informed by the collective experiences of these clinicians and researchers in the field of childhood weight management and through addressing gaps in the current literature.

### Online survey

The online survey comprised 10 questions relating to tier 2 weight management services commissioned by the LA for overweight or obese children aged 4–16 years in April 2014–March 2015. The survey collected data on the evidence base supporting the commissioned intervention, the cost of the service, the maximum number of participants the service could have accommodated, the number of children referred, the number of children completing the intervention, whether a service evaluation had been conducted and the changes in weight status measured through service evaluation. Data were collected between February and May 2016 and analysed in Microsoft Excel.

### Telephone interviews

Semistructured telephone interviews were conducted by RM between April and June 2016. The interview guide had a common framework but was adapted during the interview as guided by participants' responses.

The interviews required participants to reflect on their experiences of tier 2 weight management services for school-aged children within their LA but was not confined to experiences within the time period specified in the online survey of April 2014–March 2015. This enabled a broader representation of experiences from interview participants. The interviews explored the nature of the

contracts commissioned by LAs and the monitoring of these services through performance management and service evaluation. Specifically, the interviews explored whether outcome data were collected, whether these data were shared, the challenges identified through monitoring processes and the perceived 'markers of success' for the service.

Interviews were audio-recorded then transcribed verbatim by Bristol Transcription Services. All interview transcripts were anonymised by AP and uploaded to NVivo V.10.0 for inductive thematic analysis.

Data were organised into codes and themes and constantly revised and reviewed by two researchers working independently (RM and AP). Once coding was complete, both researchers discussed differences and links within and across themes before agreeing on the final themes. Themes were inductively and deductively elicited based on the interview guide and the information that emerged during the interviews. Data saturation was deemed to have been met when no new information emerged from the interviews, which resulted in a sample of 20 participants.[20]

### Transparency statement

The online survey was conducted as originally planned. The telephone interviews initially aimed to explore service evaluation and performance management of tier 2 weight management services for children from a commissioner's perspective and experiences. As it emerged that some LAs run inhouse contracts, participants were included who were within an LA but also service providers. The data which subsequently emerged focused the analysis on determining whether lifestyle weight management programmes were a good use of limited resources.

## RESULTS

### Quantitative data from online survey

#### Survey respondents

Contact details for 103 LA 'obesity leads' were obtained through the Directors of Public Health (DPH) and via suggestions from PHE. Of these, 40 completed the survey, 24 declined to complete the survey and provided a reason (nil commissioned n=14, service decommissioned n=4, insufficient resources to complete the survey n=3, declined for other reasons n=3), and 39 did not complete the survey and did not provide a reason. Of the

remaining 49 LAs, it is possible that the DPH forwarded our email onto the relevant contact but did not copy us in or that some of these LAs simply did not commission a tier 2 weight management service for children.

### Geographical location of survey respondents

The geographical location of the 40 LAs who completed the online survey were North West (n=10), North East (n=2), Yorkshire and the Humber (n=4), West Midlands (n=3), East Midlands (n=1), East of England (n=3), London (n=7), South West (n=7) and South East (n=3). The population of children aged 4–16 years within each of these 40 LAs ranged from 16 000 to 186 000 (Mid-2014 Population Data from Office for National Statistics).

### Evidence base of tier 2 weight management service commissioned

No LAs were aware of evidence published in peer-reviewed journals demonstrating that their service was effective at improving BMI centile (or other weight-related measure). Service evaluations were conducted in 55% of LAs, of which 18% did not measure change in weight status as part of their service evaluation. Due to heterogeneity in the way in which outcome data for change in weight status were reported by LAs (eg, proportion who reduced or maintained their BMI z-score, number who 'lost weight', % of children who reduced their BMI z-score by 3%, only 6-month or 12-month data), it was not possible to make any meaningful interpretations or comparisons of these data.

### Costs and reach of the service

Table 1 summarises the costs of the service. Some LAs were only able to provide estimates. Table 2 summarises the reach of services within an LA.

### Qualitative data from telephone interviews

Twenty telephone interviews were conducted with LAs (18 commissioners, 2 service providers within the LA—interview numbers 18 and 20). Seventeen of the telephone interview participants had completed the online survey. Three had declined. The geographical location of the 20 LAs who completed the interview were North West (n=8), North East (n=1), Yorkshire and the Humber (n=1), West Midlands (n=0), East Midlands (n=1), East of England (n=1), London (n=4), South West (n=3) and South East (n=1). Interviews were between 28 and 68 min in length.

| Table 1 Costs of the service | |
| --- | --- |
| | Mean cost (SD, n) |
| Cost of the service per year to LA | £130 742 (£122 869, n=27) |
| Cost of the service per year per 10 000 children aged 4–16 years (of any weight) in LA | £29 397 (£30 003, n=27) |
| Cost of the service per overweight or obese child attending if maximum capacity of the service was reached | £558 (£408, n=18) |
| Cost of the service per child completing the intervention | £1312 (£1342, n=15) |

LA, local authorities; n, number of LAs providing data.

| Table 2 Reach of the service | |
|---|---|
| | **Mean (SD, n)** |
| Potential reach of the service (presuming maximum capacity was achieved) to overweight or obese children within an LA | 3.5% (6.9%, n=26)* |
| Estimated actual reach of the service (ie, children completing the intervention) to overweight or obese children within an LA | 1.2% (1.6%, n=25)* |

*These calculations used estimates of the prevalence of overweight or obese children within an LA aged 4–16 years (this was estimated using NCMP data from reception and year 6 and National Statistics population data for children aged 4–16 years).

LA, local authority; NCMP, National Child Measurement Programme.

## Nature of commissioning contracts

Tier 2 weight management contracts were either between the LA and an external provider, or 'in-house' contracts (where the LA acts as both the commissioner and the service provider). Some LAs reported running 'in-house' contracts as they could not afford to commission the service to an external provider. This was not a problem if the service was performing well; however, if the service was underperforming, their options might be limited as they may not be able to go out to market due to financial and political pressures.

> if they're not achieving their targets, they're not doing their job properly, so then we shouldn't be providing the service, but what is the alternative? It's too expensive to commission it out. (INT 3)

One LA discussed the challenges of 'in-house' contracts from a leadership perspective, especially as their service was not meeting BMI targets.

> To make it complicated our provider is also within the LA so there's a bit of – it's something that provides such a huge challenge just on its own because you've got provider senior leadership and commissioning senior leadership with different views…the service underachieved against the targets around BMI consistently over the last two years…If they were an external provider it would probably be a different scenario. (INT 17)

## Outcome data

All LAs collected outcome data through performance management processes and some also collected outcome data through service evaluation. Most interventions were around 12 weeks long, with data collected at baseline and at the end of the intervention. Some LAs also attempted to collect longer-term data at 3 months, 6 months and/or 12 months. Although the general themes of data collected were similar (demographic data, retention, engagement, weight, self-esteem, confidence, behavioural change, physical activity, diet), the actual data were collected in different formats across some LAs. For example, some LAs measured physical activity via a 7-day recall questionnaire, others through a physical activity test and others by asking parents whether their children increased their activity levels or through assessing physical literacy.

## Challenges identified through service evaluation and/or performance management meetings

### Lack of long-term data

Many participants mentioned the difficulties in collecting long-term follow-up data. This was attributed to a variety of factors, including length of questionnaires, lack of parental confidence with the paperwork, too much effort for families to undertake, people moving around town, resource constraints of LA to capturing these data, lack of information technology infrastructure, and lack of engagement in both the intervention and the evaluation.

> It becomes then quite time consuming to try and chase patients who engaged. People forget what they've done 12 months ago or more as well. …it would be quite difficult with not having things like a GP (General Practice) surgeries infrastructure like EMIS (Egton Medical Information Systems) where data gets held for years and years and it's there to use and accessed again. (INT 14)

### Lack of validated tools

Some participants felt that there was a lack of validated tools to enable accurate outcome measures to be obtained.

> We're looking for validated tools but there are just not that many great ones out there. (INT 1)

### Reliability of self-report data

A few participants questioned the reliability of self-report data.

> Other challenges are self-reporting. …The physical activity and nutrition tend to be improved after ten weeks and sometimes you look on that a little cynically because the measurements haven't improved, so perhaps they're telling us what we want to hear, that can be a challenge. (INT 3)

### Lack of engagement

Difficulties engaging children, parents and healthcare professionals with the service were mentioned by many of the LAs. These are summarised in table 3.

### Lack of resources/expertise

A few commissioners felt that service providers lacked expertise in conducting service evaluations.

**Table 3** Challenges of engaging parents, children and healthcare professionals with the service

| Difficulties engaging parents | |
|---|---|
| Talking about the weight of a child can be highly emotive for parents. | "It's difficult with parents sometimes to explain to them that what they are doing at home is probably not the best thing for their child. That's quite difficult you know, that's their baby that's their child and they don't want to hear anything negative." (INT 19) |
| Parents often find it difficult to accept that their child is overweight. | "Parents often see their children as normal weight when they are in fact overweight and we know people often will refer to children who are a normal weight as a bit skinny." (INT 5) |
| Parents often do not recognise the role they need to play in engaging in the service as part of a 'family intervention' to improve their child's BMI centile. | "So we say it has to be a family intervention. But they don't always see it that way. They just want the child to lose the weight and don't acknowledge their role in being the providers' food and the environment they grow up in." (INT 11) |
| Difficulties engaging children | |
| Engaging children with the service could be challenging. | "there is a lot of issues around recruitment and retentions with tier two services for children and also there's a great difficulty with actually the secondary aged children to get them sort of accessing services." (INT 2) |
| Difficulties engaging healthcare professionals | |
| Healthcare professionals can find it difficult to bring up weight status of a child with a parent. | "I think there's definitely issues there from what I've heard about professionals bringing things up with families." (INT 4) |
| Some healthcare professionals fail to recognise overweight or obese children. | "The GP will look at the child and say, it's just puppy fat, they'll grow out of it." (INT 5) "We even get some out of school nurses say 'well, they're only just into the overweight category'. You know, the child is really athletic, they're really muscular." (INT 11) |
| Lack of GP engagement. | "GP's still struggle to engage with it." (INT 10) "GPs, locally they tend not to refer." (INT 6) |

BMI, body mass index; GP, general practitioner.

there's difficulties there with the data that we need because we also find that the skill set of a lot of the people delivering the services doesn't always sit with evaluation. (INT 14)

### Financial pressures on services

There are considerable financial pressures facing LAs at present, and budget constraints are impacting on the provision of tier 2 weight management services for children in most LAs in different ways.

we're at a point now where we're going through council budget savings, the service has actually taken a 50% hit, which is huge…so how are we supposed to achieve this whole you know like city wide target on less money is going to be impossible…We've got smaller and smaller services and you keep telling me you're going to take some more money away from me so how are we supposed to achieve these things. (INT 17)

Some LAs have found it challenging to provide a good service with reduced funding. Strategies taken to cope with the funding cuts have included setting lower targets as part of the key performance indicators (KPIs).

we've had to work together to reduce the KPIs anyway because they just wouldn't be met with that much of a dent in the finances. (INT 1)

A few LAs are considering, or have already decided, to decommission their weight management service.

So yeah things are really tight and at the regional network meeting people were talking that they may have to de-commission their weight management services. (INT 2)

LAs talked about the need to demonstrate 'good value for money' for a service to justify its funding.

I'm constantly looking at a cost benefit analysis and working out, okay how much is this costing per child, how much is it costing per family? What are the outcomes that we're getting? Is this really a programme that is cost effective? (INT 5)

A few LAs discussed the difficulties in allocating money to service evaluation when money for service provision itself was so limited.

### Pressures on service to influence the prevalence of obesity

LAs often described the pressures they are under to reduce the prevalence of obesity within their borough through their tier 2 weight management programme. In some LAs, this seemed to be politically driven by councillors.

They're fixated about our actual prevalence rate… the councillors yeah and sort of senior management. We've got like sort of corporate score card and they

wanted to put obesity prevalence as part of that. (INT 2)

Reducing the prevalence of obesity was frequently seen as an unrealistic goal given the reach of the service often being so small, the funding allocated limited and the feeling that one service cannot be accountable for solely tackling such a complex problem with a programme length that is usually only 10–12 weeks.

In terms of tackling childhood obesity I'd say the child weight management programmes are family weight management programmes, they're only going to go so far. We know our population in LA14, we've probably got 500 families within each year group that would be affected by obesity even more that would be affected by overweight. If you times that by 18 years of childhood you've got quite a significant number of families up in the 10 000 maybe that are going to have these weight management issues. We're never going to be able to commission a service that would be able to work at a one to one level or a group level with 10 000 families, it's not going to be practical to do that. On the other side of things, we're looking at strategies that take a much more preventative approach. (INT 14)

To achieve the objective of reducing the prevalence of obesity, some LAs recognised that population-based approaches would be required.

the number of people we're getting to is actually quite small…it's not going to change obesity levels locally, so we do need to look at more population-based approaches so that's something we will be doing…I suppose doing less programmes possibly in future because the numbers per programme aren't as high as we'd want. (INT 15)

### Need for a 'Whole Systems Approach'

Many of the LAs talked about a recent shift towards a 'whole systems approach' to tackling obesity[21–23] and the need to view weight management schemes alongside the 'bigger picture'.

we can run weight management schemes and I think they're really important, but it has to be part of the bigger picture because you know children's families only go to those sort of schemes like once a week. It's their whole environment that it's important to actually help them to making behaviour change and actually if we don't do both and try and change the obesogenic environment people aren't going to be successful in weight management and it's only going to be a short term, isn't it. (INT 2)

Some felt that national strategies to try and change the obesogenic environment (eg, active transport, sugar tax, change for life campaign) and perception of what constitutes a healthy weight were needed to influence the prevalence of obesity.

It's not going to be easy because it's more and more difficult to make healthy living the norm because it's just too easy to be unhealthy. It's going to take a major upheaval for it to get any better. I think the sugar taxes could help, I think we're going to see more and more of these. What I think we could do to improve it is more and more national campaigns, that's what I think. (INT 3)

### Sharing and use of evaluation data

Most LAs showed willingness to share data; however, this tended to happen on an informal 'when requested' basis. Some LAs reported sharing data with other LAs more formally through obesity network meetings or emails, but this was at a regional rather than national level. Suggestions for future sharing mainly focused on developing online networks, forums or webinars which would enable data to be accessed both at a regional and national level.

I know in the sexual health areas they have like a forum or something, a website and they all sort of meet up and share best practice and they can ask questions online and things like that, so something like that for weight management would be good. (INT 13)

I think there could be like a national monitoring… It would be useful to be able to know exactly what data is needed and have methods for having that all collected in one place by one system and then to be able to pull reports from that system locally, regionally, sub-regionally, nationally and even if we could go down to a very local level even a ward level. (INT 14)

Some LAs felt that regional and national child obesity commissioner meetings would be useful. A few barriers mentioned to sharing data included time pressures, the commercially sensitive nature of some information and potential competition between LAs, although most did not feel that the latter was a significant issue.

Within LAs, evaluation or performance management data were mainly used to reshape and improve services and sometimes to promote the service and secure future funding.

### Future directions
#### *Guidance needed on service specifications and contracts*

Many LAs commented on the lack of consistency in service provider contracts, specifications and outcomes measured across different LAs. They felt that detailed practical guidance with sample service specifications and service provider contracts would be useful, including detailed guidance on what exactly the service should be aiming for in terms of weight loss and other objectives.

I mean there's no like commissioning guidance on weight management programmes you know if that appears on my desk I'd be a very happy bunny because

you know then it will tell me exactly what I need to look for, exactly what needs to be achieved. But we don't have a guidance that tells us that you know this is what you should expect from your provider. (INT 19)

I know trying to find some sort of consistency I think from a contracts point of view, it's been helpful that in other services, not children's weight management where we have had collaborative working around specifications and contracts and then obviously their local detail has been added to it. (INT 17)

### Cost-benefit analyses tool

In the current economic climate, a few LAs suggested that it would be helpful if researchers developed a cost-benefit analysis tool which they could use for their child weight management programmes to justify allocation of money to these programmes.

a cost analysis tool. So, in terms of if X loses 5% in terms of weight loss, what that saves NHS/CCG/whoever it may be long term, because we have these cost analysis tools for *another service within the LA*, we have GP cost per hour, things like that, but we don't have anything for weight management for young people, but a cost analysis tool would be great. (INT 18)

### DISCUSSION
### Main findings

Data from the online survey demonstrated that no LAs were aware of any peer-reviewed evidence supporting the effectiveness of their specific tier 2 weight management service at improving BMI centile. Service evaluations were not consistently conducted. There was little consistency in methods for reporting change in weight status. The mean cost of the service per child completing the intervention was £1312, and the mean actual reach of the service (ie, children completing the intervention) to overweight or obese children within an LA was only 1.2%.

The qualitative research revealed the complexities of 'in-house' contracts in some LAs. There were similarities between LAs in the length of the intervention programme commissioned, the timing of data collection points and the outcomes measured. There were inconsistencies in the tools used to measure these outcomes, which complicates meaningful comparisons of data between LAs. Formal sharing of data between LAs was lacking. LAs identified many challenges facing their service in both provision, through lack of engagement and lack of resources, and inservice evaluation, through the questionable reliability of self-report data, lack of validated tools and difficulties in collecting long-term data.

Many LAs described the pressure on their service to reduce the prevalence of obesity but felt that a 'whole systems approach' was needed to tackle this problem rather than over-reliance on a single service. Some LAs felt more detailed guidance was needed on service

specifications and contracts. Development of a cost-benefit analysis tool was also discussed by a few LAs.

### Meaning of the findings: implications for policy makers and clinicians

There is currently no way of easily comparing BMI z-score or other outcome data between different tier 2 weight management programmes across multiple LAs in England. Although PHE has recently developed data entry forms, there is no mandatory system in place requiring LAs to submit this information so it can be collated onto a central database for analysis.[24] Where data are shared, this is usually done on an informal basis at a local level. This is surprising given that the online survey highlighted that no LAs knew of any peer-reviewed evidence supporting the effectiveness of their service at influencing weight status. In addition, there are very few UK-based research trials demonstrating a clinically significant reduction in BMI z-score in school-aged children (defined as mean BMI SDS reduction of ≥0.25).[25–27] A recent systematic review by Burchett et al[26] reported only 5 of the 30 interventions included in the review reduced BMI z-score by ≥0.25. Of these five interventions, none was conducted in the UK and only one involved children of school age.

Given the current economic climate and lack of evidence regarding long-term effectiveness of these interventions, it would seem wise to ensure that outcome data were being collected in a standardised format and that these data were compared and shared. This could help local and national agencies such as PHE to make evidence-based, cost-effective commissioning decisions, as the data in this paper suggest that these decisions are currently being conducted without good-quality evidence of long-term benefit. However, even if this was achieved, many LAs have already alluded to the difficulties in collecting long-term data and so it is likely that there would be important gaps. It is also plausible that where long-term data are collected, no long-term effectiveness is demonstrated. This is possible given that the NICE evidence review supporting the PH47 guideline reported no statistically significant mean difference in BMI z-score in the long-term for lifestyle weight management interventions for children.[17]

Many LAs discussed the pressures on their service to reduce the prevalence of obesity. However, the actual mean reach of a service (ie, children completing the intervention) to overweight or obese children within an LA was 1.2%. It is therefore unrealistic to expect these services to influence obesity prevalence rates. Population measures are needed to have population-level effects, and it is therefore unclear where tier 2 services such as those evaluated fall within the overall obesity strategy as they are neither population-focused nor have a strong evidence base for clinically defined groups. Even if the service had the capacity to take a large proportion of the overweight or obese population, the programme still probably would not reach most of this population due to the difficulties in engagement discussed by LAs in the telephone interviews. Problems engaging families with services have been recognised in

the literature.[28] Many LAs described the need for a 'whole-systems approach' to effectively tackle the problem of childhood obesity.

A whole systems approach recognises the need to address a complex multicausal problem using multiple different approaches rather than through a single intervention alone.[29–31] At an LA level, this may involve influencing and linking multiple sectors such as planning, housing and transport to effect population-level changes.[30 31] Allender *et al*[32] describe a community's understanding of the complex causality of obesity through a causal loop diagram, and they outline an obesity prevention trial aiming to use a whole systems community-led approach.[33] PHE has commissioned Leeds Beckett University to identify ways in which LAs might achieve a successful whole systems approach.[31]

### Weaknesses

The sample size for the online survey and telephone interviews was relatively small, but there was good geographical representation across England and saturation was felt to have been achieved in the telephone interviews. It is also not mandatory for LAs to commission a tier 2 weight management service, so some LAs may have felt this research was irrelevant. Due to the method of recruitment to our study, it is possible that in some LAs, details regarding the online survey did not reach the relevant person. A freedom of information request was not submitted to obtain missing data and this is a limitation of the study. No implementation theories were used to evaluate programmes.

Although a topic guide was used for the interviews, further discussions were guided by the participant. This had the strength of allowing inductive analyses to be conducted, but the weakness that the opinions of every interview participant on each of the themes reported may not have been captured. It is also important to note that the current study focused on LAs in England. This means that the generalisation of results to the rest of the UK and wider is unclear.

Finally, LAs did not provide answers in a comparable format for all questions on the online survey, which limited statistical analyses to a relatively small number of LAs. This was likely in part due to variation in the type and format of data collected by each LA. A recent PHE study also recognised this problem, reporting that the average change in BMI centile postprogramme and at 12 months could not be determined due to the heterogeneity of respondents.[34] To gain a true oversight of the cost-effectiveness of lifestyle weight management programmes currently commissioned in the UK, there needs to be consistency in the outcomes measured and clear guidelines on what clinically significant outcomes these services should be aiming to achieve.

### Strengths and contextualisation

In 2015, PHE conducted a national mapping study of tier 2 and tier 3 weight management services.[34] The evidence base for the commissioned service was determined by asking LAs if they followed NICE guidance or not, rather than asking whether their commissioned service had evidence supporting effectiveness in the peer-reviewed literature, as in this study. This is an important distinction as using guidelines to facilitate commissioning decisions is different from demonstrating the effectiveness of a commissioned service, especially given the limitations of the evidence supporting the NICE PH47 guidelines.[35 36]

The PHE mapping study stated that the most frequently reported cost per participant of the service was ≥£401, although there is no further breakdown on figures above this range nor any SD or mean costs provided. As a result, it is not possible to estimate the cost-effectiveness of these interventions. Furthermore, the survey asks for the 'average cost of the intervention per participant' but does not specify whether this should be per participant referred, per participant starting the intervention or per participant completing the intervention.[37] The strength of our study is that we distinguish costs between these groups and report their means (with SD).

In order to determine whether participants are followed up long term, the PHE mapping study asked: 'How long are the providers required to follow up the participants?' The study reported that 67% of services reported follow-up of participants for 12 months or more. However, being 'required' to follow-up does not mean that the data were collected for all these participants. Our qualitative data provide insight into the difficulties in collecting long-term data even when the specification to do so is present.

The qualitative aspect of the PHE study had some similarities with our research, reporting lack of evidence of long-term effectiveness, lack of validated tools, lack of clear guidance on specifications, lack of funding, lack of expertise and difficulties with recruitment.

### Future directions

In their present format, tier 2 weight management services for overweight and obese children are very unlikely to have any impact on the prevalence of childhood obesity, and peer-reviewed evidence of any long-term benefits even for the small numbers of children reached by these services is weak. If these lifestyle weight management services are to be continued, clear thought needs to be given to the goals of the service, and a more robust independent system needs to be developed to determine whether these goals are being met, whether the service is cost-effective and if it is the best use of limited resources in the current economic climate. Subsequently, if cost-effectiveness is demonstrated, work needs to be undertaken to understand the variation in the provision of these services across England, such as through an 'Atlas of Variation[38]', and how LAs can be supported in the commissioning and delivery of these services, given that they are non-mandatory.

However, it is also important to consider whether preferential investment should be given to population-level approaches or to developing strategies to deliver a whole

systems approach by LAs rather than investing in a single small-scale, lifestyle weight management programme. Population measures such as the sugar tax have been identified as having the potential to reduce the prevalence of obesity, with the greatest benefit predicted for those under the age of 18.[39] In Mexico, the tax on sugar sweetened beverages (SSBs) in 2013 was associated with fewer taxed beverages being bought and more untaxed beverages being bought.[40] A similar tax in California reduced SSB consumption in low-income neighbourhoods.[41] Yet there is limited direct evidence of a link between a sugar tax and reduction in obesity prevalence aside from modelling studies. Other population-level strategies include reduction of television advertising of high-fat and/or high-sugar foods and drinks to children,[42] nutritional labelling of foods, transport policies and multicomponent mass media campaigns.[43] Nonetheless, Dobbs et al suggest that public health campaigns have the least evidence for cost-effectiveness.[44]

Regardless of how funding is allocated to tackling obesity, there needs to be robust cost-effectiveness analyses and sharing of data nationally to help inform future commissioning decisions and ensure that scarce financial resources are being used in the most efficient and effective way across England.

## CONCLUSION

Our results show that none of the participating LAs was aware of any peer-reviewed evidence supporting the effectiveness of their specific commissioned service. Despite this, there was no national formal sharing of data to enable oversight of the effectiveness of commissioned services across LAs in England to help inform future commissioning decisions. Challenges with long-term data collection, service engagement, funding and the pressure to reduce the prevalence of obesity were frequently mentioned. The need for a 'whole-systems approach' to tackle obesity effectively was discussed. In the future, obesity treatment or prevention programmes need to have robust systems in place to feedback programme outcomes and costs in a comparable and transparent format to enable national, independent oversight of the cost-effectiveness of different obesity strategies and direct future commissioning decisions.

**Author affiliations**
[1]Centre for Academic Primary Care, Bristol Medical School, University of Bristol, Bristol, UK
[2]Centre for Exercise, Nutrition and Health Sciences, School for Policy Studies, University of Bristol, Bristol, UK
[3]Department of Population Health Sciences, Bristol Medical School, University of Bristol, Bristol, UK
[4]Department of Population Health Sciences, Centre for the Development and Evaluation of Complex Interventions for Public Health Improvement (DECIPHer), Bristol, UK
[5]NIHR Bristol Biomedical Research Centre, Nutrition Theme, University of Bristol, Bristol, UK
[6]Faculty of Health Sciences, University of Bristol, Bristol, UK

**Contributors** JPHS, RJ, DS, RK and RM conceived and designed the study. Provisional analyses of the online survey data were conducted by RM and the telephone interview data by RM and AP. The results and analyses were reviewed and discussed by JPHS, RJ and DS. RM drafted the manuscript. All authors revised and approved the final manuscript. The corresponding author attests that all listed authors meet the authorship criteria and that no others meeting the criteria have been omitted. RM is the guarantor.

**Funding** RM is funded by the NIHR (NIHR ACF 2014–2017, NIHR In-Practice Fellowship 2018–2021). RK works at the Centre for the Development and Evaluation of Complex Interventions for Public Health Improvement (DECIPHer), a UKCRC Public Health Research Centre of Excellence: joint funding (MR/K0232331/1) from the British Heart Foundation, Cancer Research UK, Economic and Social Research Council, Medical Research Council, the Welsh Government and the Wellcome Trust, under the auspices of the UK Clinical Research Collaboration. JPHS is supported by the National Institute of Health Research (NIHR) Biomedical Research Centre at the University Hospitals Bristol National Health Service (NHS) Foundation Trust and the University of Bristol, Bristol, UK. DS works at CAPC PHS funded by HEE Severn. RM is funded by a National Institute for Health Research (NIHR) Academic Clinical Fellow (ACF-2012-25-009) and In-Practice Fellowship Award (NIHR-IPF-16-10-07) for this research project.

**Competing interests** RM and JPHS report grants from the National Institute for Health Research. RJ reports grants from NIHR, and RK reports grants from DECIPHer and grant funding from a collection of charities and funding councils as outlined in the Funding section of this paper.

**Patient consent for publication** Not required.

**Ethics approval** Ethical approval was granted by the School for Policy Studies Research Ethics Committee at the University of Bristol. Informed consent was obtained in written format for the online survey and in written or verbal format for the telephone interview.

**Provenance and peer review** Not commissioned; externally peer reviewed.

**Data availability statement** All data relevant to the study are included in the article or uploaded as supplementary information.

**ORCID iDs**
Ruth Mears http://orcid.org/0000-0002-1498-6996
Russ Jago http://orcid.org/0000-0002-3394-0176

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
