## [Reviewer comments · BMJ Open]

ARTICLE DETAILS

TITLE (PROVISIONAL)	A mixed methodology study exploring how lifestyle weight management programmes for children are commissioned and evaluated in England.
AUTHORS	Mears, Ruth; Jago, Russ; Sharp, Deborah; Patel, Anamica; Kipping, Ruth; Shield, Julian PH

VERSION 1 – REVIEW

REVIEWER	Dr Matthew S Capehorn Rotherham Institute for Obesity (RIO) United Kingdom
REVIEW RETURNED	06-Sep-2018

GENERAL COMMENTS	This is a valuable paper that highlights the deficiencies in commissioning childhood weight management services, and it will be a useful addition to the evidence base that is required to highlight changes that are needed in terms of choice of provider, data collection and service evaluation etc. However, I do not feel that the findings justify the proposed conclusion that preferential funding should be given to population-level based interventions rather than tier 2 weight management services. Based on the premise within this paper that there is limited evidence for the effectiveness of these T2 interventions, there is actually no robust evidence for population-level interventions either. At most the conclusion must be that equal emphasis should be given to prevention and treatment, but to engage with arguments about preference of limited resources, then further discussion would be required based on the other reports and evidence that have looked in to this including the McKinsey Report of 2014 that suggested population-level public health campaigns had little evidence and the least amount of impact of obesity, especially when compared with the evidence for weight management interventions (although this will have included tier 3 and tier 4 interventions also). However, I accept that the conclusion states consideration should be given to this rather than proposing this, but I feel the way it is wording may be trying to influence the reader. In terms of the methodology in this paper, it accepts the limitations, but is not explicit in who completed the survey or the telephone consultations, and whoever completed these will have naturally skewed the responses. It states that the Dir of PH for each LA was asked as to who the lead was. Do we have a breakdown of how many were able to name a LA or PH lead for obesity, as more recent study based on FOI data have shown that many LAs did not know who was responsible? What was the actual percentage who could name a lead, and what percentage of these leads were the people who completed the surveys/consultations, as it later
--

suggests that in a couple of occasions service providers were consulted. All of this questions the reliability, credibility and bias within the answers given (although some of the themes appear to be quite consistent).

What percentage did not even respond to the online survey and why was a FOI request not submitted for the relevant information for this part of the study? Do we have any data listing reasons for not completing either part of the study, or is it assumed that if they did not then this was not followed up, and assumed to be due to apathy or no commissioned service. A chart breaking down the responses more clearly would have been helpful. Were any of the negative responses (ie either no response, or a response stating no commissioned service) because their local service catered for a different age group to that being studied, ie not 4-16years, as some areas may have had services available for just older children or perhaps even younger children. Did respondents incorrectly assume that if they did not cater for this whole age range then they could not respond. Subsequent FOI surveys have identified that respondents often make mistakes in the interpretation of the questions, and without follow up telephone calls to clarify then mistakes are made.

Also, do we now how many of these tier 2 services work in collaboration with a commissioned Tier 3 service (whether commissioned by the same LA or local CCG) and did this influence response rates or answers given? The fact that responses could be given in relation to a locally commissioned service even if it was outside of the time period being assessed in the study, may imply some responses were retrospective and related to services that were subsequently decommissioned. This might suggest a higher response rate than reflects activity, and actual figures are worse than presented. I am not sure whether it is a valid statistical method to determine the final sample size by saturation of information, ie when no new information seemed to emerge, giving a final sample size of 20.

The results suggest that 64 LAs responded. Was this 64 x Dir of PH responded to the initial contact, or 64 responses from nominated obesity leads? If so, why did only 40 respond to the online survey? An FOI request would have gathered valuable information. The fact that none of those completing the online survey knew of any evidence published in peer-reviewed journals demonstrating that their service was effective at improving BMI centile is very significant. However, it is worth noting that this is not the same as assuming that this evidence is not their, as there is published data on the success of many child based T2 interventions, such as MEND, More Life, SHINE, etc.

The cost data for each of the services is very interesting and valuable data, that I have not seen presented before. It would have been useful to see whether services that worked in collaboration with an established childhood tier 3 service (whether community based or secondary care) spent more or less than average on their tier 2 service.

It is not clear from the text whether the 20 respondents to the telephone interviews were the named PH obesity lead or other. It does state that at least 2 were not, as they were service providers. I feel that the role of the respondent may well have given differing responses to similar questions, that was not based on differences in the services.

The fact that this highlights a consistent theme where commissioners have no standardised format for outcome data is significant and should be highlighted more in order to influence

	future commissioning. However, other responses need to be contextualised as we may be getting the opinions of non-experts in the field and we know that there are emotive and political agendas when it comes to childhood obesity. I think the section on future directions was good, with specific reference to the need for guidance on service specifications, contracts, collecting data, and analysing cost-benefit. However, more could have been made about the fact that childhood obesity services are not mandated, and as such the LGA have already stated that in times of financial restraint any non-mandated services, like obesity, will not be protected. There should have been discussion about the need for services to be mandated, along with a minimum level of service provision to complement the future directions suggested, together with the continuation of ring-fenced funding. In the discussion the authors make the observation that the reach of the services was only very small at 1.2%, however without giving population level context. ie data produced by More Life has previously demonstrated that for the large majority (62-98%) of obese adolescents, tracked from those children who were obese at a young age, ie relatively few healthy weight children made up the total of obese adolescents. So to save valuable resources (and by no means dismissing the importance of population-level public health campaigns for the masses) these young obese children need to be targeted, and so if he reach of 1.2% specifically targets a significant proportion of those obese children at a young age then it may have more long term impact than suggested. The lack of reach and lack of evidence for effectiveness is argued against the value of population level interventions such as the sugar taxes introduced in other countries. However, I feel that if the sugar tax is being proposed as an example of a public health intervention then a balanced argument should be given, exploring the limited benefits seen as a result of these taxes in countries that have introduced it and the potential disadvantages of this approach, otherwise again it is trying to influence the reader into preferring the public health approach. The WHO is also stated as having proposed other interventions for preventing childhood obesity, but the evidence base for these proposals are far from robust, without acknowledging this.
--	--

REVIEWER	Miranda Pallan University of Birmingham I am collaborating on a research funding application with two of the Authors (Ruth Kipping and Russ Jago). The proposed research is not related to the provision of children's weight management services.
REVIEW RETURNED	10-Sep-2018

GENERAL COMMENTS	This is a very well written paper which contains information of interest to public health practitioners and healthcare commissioners in England. I have a few amendments and suggestions to improve the paper further:  1. Introduction, end of first paragraph and second paragraph – provide some explanation of what CCGs, NHS England and Public Health England are for international readers
--

	2. Introduction, 4th paragraph – it would be helpful to clarify what you mean by evidence-base here (e.g. evidence underpinning the service provided in the local area) 3. Methods, online survey section – Please include more information on how this was developed – how did you determine the questions to include? 4. Methods, telephone interviews – Again, more detail on how the interview guides were developed, and what informed this process would be useful 5. Results, Evidence-base for Tier 2 weight management service commissioned – it would be helpful to include examples of differences in outcome data collected by LAs 6. Results, Table 1 – the standard deviations given are very big and seem implausible (i.e. bigger than the mean in some cases); please check your figures for this 7. Results, Table 1 – in the foot note, please clarify that you are referring to the denominator used 8. Qualitative data – for quotes given, please state if they came from a commissioner or provider 9. Results, outcome data - Give the numbers of LAs that collected data through performance management and those that collected data through evaluation. Also make it clear whether the data comes from the quantitative survey or the qualitative interviews 10. Results, outcome data – the authors give examples of outcome measures that were unique to one or two LAs, from my reading it seems as if these probably relate to specific programme content. If this is the case, then it would be worth stating it in the manuscript 11. Results, whole systems approach – this terms can mean a whole variety of things, it would be worth clarifying what the Local Authority representatives interviewed meant by this 12. Discussion, 2nd paragraph – I would recommend separating the challenges associated with service provision and those associated with service evaluation 13. Discussion, Meaning of the findings: implications for policy makers and clinicians, last paragraph – there is international literature on the use of whole systems approaches for obesity prevention, which could be incorporated here (see work done in Australia by Steve Allender and colleagues) 14. Discussion, strengths and contextualisation, first paragraph – reference the evidence supporting NICE PH47 (links to the systematic reviews informing the guideline are available on the NICE website).
--	--

REVIEWER	Meliha Salahuddin University of Texas Health Science Center at Tyler, Texas, USA
REVIEW RETURNED	26-Dec-2018

GENERAL COMMENTS	General comments  • Please be consistent in the usage of “LA.” The authors shift between the full and abbreviated form throughout the manuscript. • Same for PHE. Introduction  • Please provide the full form for PH47 (page 3 line 32) and SMD (page 3 line 34). • Please clarify what is “ring-fenced public health budget” (page 3 line 46). Methods  • Please clarify whether any implementation science theories were used to evaluate the program such as RE-AIM framework (http://www.re-aim.org/), CFIR (https://cfirguide.org/constructs/), as examples. The authors may wish to contextualize their approach/analysis using these frameworks. The authors may find the following papers useful in determining the appropriate constructs along with the theory:  1. Proctor E, Silmere H, Raghavan R, et al. Outcomes for Implementation Research: Conceptual Distinctions, Measurement Challenges, and Research Agenda. Administration and Policy in Mental Health and Mental Health Services Research. 2011;38(2):65-76. doi:10.1007/s10488-010-0319-7 2. Rogers EM. Diffusion of innovations. 4th ed. New York: The Free Press; 1995. 3. Glasgow RE. Translating Research to Practice. Lessons learned, areas for improvement, and future directions. 2003;26(8):2451-6. doi:10.2337/diacare.26.8.2451 4. Glasgow RE, Vogt TM, Boles SM. Evaluating the public health impact of health promotion interventions: the RE-AIM framework. American Journal of Public Health. 1999;89(9):1322-7. doi:10.2105/ajph.89.9.1322 • Were any psychometric analyses done to evaluate the measures (e.g., reach)? Psychometric analysis would increase the methodological rigor of the study. Furthermore, could the measures be combined to form an index (Salahuddin, Barlow, Pont, Butte, & Hoelscher, 2018)? • Sampling technique: The sampling technique for the semi-structured interview was not clear – it sounded arbitrary based on the comment “the final sample size was determined by the saturation of information when no new information seemed to emerge. This resulted in a final sample of 20 participants.” Was any power analysis conducted to determine the sample size? (page 6 lines 20-
--

	21). Why did the authors not conduct the semi-structured interview with all 40 participants that completed the online survey?  Here the authors are comparing 40 different LA commissioned tier 2 weight management services. It is not clear what the programs consisted of and whether they were all similar. Providing more information about the different program components is important. Otherwise, it is not clear whether the authors are comparing apples to oranges. Furthermore, the age range is broad (4-16 years). I would imagine weight management programs would be different for a 4-5-year-old versus a teenager. Additionally, it would be different depending on the severity of obesity (children with severe obesity (%BMIp95 ≥ 120) versus children with obesity (%BMIp95 < 120)) Results  It is a critical limitation that the different weight-related outcomes measures were not evaluated as different programs measure different outcomes (page 7 lines 32-36). It is not clear what is the difference between the first two rows in Table 1. Discussion  The first paragraph of the Discussion typically includes a summary of the main findings. It is well-documented in the literature that when it comes to obesity prevention, a holistic approach involving multiple sectors and multiple levels (individual, family, school, community) is effective. It is great that the local authorities identified this as a potential next step. Additionally, population-level intervention and measures would be more effective and particularly, intervening at an early age as obesity is known to track into adulthood. Reference: Salahuddin, M., Barlow, S. E., Pont, S. J., Butte, N. F., & Hoelscher, D. M. (2018). Development and use of an index for measuring implementation of a weight management program in children in primary care clinics in Texas. BMC Family Practice, 19(1), 191. doi:10.1186/s12875-018-0882-7
--	--

VERSION 1 – AUTHOR RESPONSE

Reviewer 1: Dr Matthew S Capehorn

Institution and Country: Rotherham Institute for Obesity (RIO), UK

Comment 1: This is a valuable paper that highlights the deficiencies in commissioning childhood weight management services, and it will be a useful addition to the evidence base that is required to highlight changes that are needed in terms of choice of provider, data collection and service evaluation etc.

Response: Thank you for your comments recognising the relevance and importance of this paper in the future commissioning and evaluation of childhood obesity services.

Comment 2: However, I do not feel that the findings justify the proposed conclusion that preferential funding should be given to population-level based interventions rather than tier 2 weight management services. Based on the premise within this paper that there is limited evidence for the effectiveness of these T2 interventions, there is actually no robust evidence for population-level interventions either. At most the conclusion must be that equal emphasis should be given to prevention and treatment, but to engage with arguments about preference of limited resources, then further discussion would be required based on the other reports and evidence that have looked in to this including the McKinsey Report of 2014 that suggested population-level public health campaigns had little evidence and the least amount of impact of obesity, especially when compared with the evidence for weight management interventions (although this will have included tier 3 and tier 4 interventions also). However, I accept that the conclusion states consideration should be given to this rather than proposing this, but I feel the way it is wording may be trying to influence the reader.

Response: The future directions and conclusion have been amended to put greater emphasis on the need for evidence-based, cost-effective strategies and standardisation of outcome data and less emphasis surrounding the approach to tackle obesity (T2 intervention vs population-level based interventions). The McKinsey Report suggests that obesity cannot be addressed by a single intervention alone but rather needs 'a systemic, sustained portfolio of initiatives, delivered at scale' and we have added the need to consider investment into developing strategies to deliver a whole systems approach by LAs. The limited cost-effectiveness evidence for public health campaigns reported by McKinsey et al has now been cited. The limitations of the sugar tax evidence to modelling studies has also now been discussed.

Comment 3: In terms of the methodology in this paper, it accepts the limitations, but is not explicit in who completed the survey or the telephone consultations, and whoever completed these will have naturally skewed the responses. It states that the Dir of PH for each LA was asked as to who the lead was. Do we have a breakdown of how many were able to name a LA or PH lead for obesity, as more recent study based on FOI data have shown that many LAs did not know who was responsible? What was the actual percentage who could name a lead, and what percentage of these leads were the people who completed the surveys/consultations, as it later suggests that in a couple of occasions service providers were consulted. All of this questions the reliability, credibility and bias within the answers given (although some of the themes appear to be quite consistent).

What percentage did not even respond to the online survey and why was a FOI request not submitted for the relevant information for this part of the study? Do we have any data listing reasons for not completing either part of the study, or is it assumed that if they did not then this was not followed up, and assumed to be due to apathy or no commissioned service. A chart breaking down the responses more clearly would have been helpful.

Response: We have amended the results section to provide further clarification;

'Contact details for 103 LA 'obesity leads' were obtained through Directors of Public Health and via suggestions from PHE. Of these, 40 completed the survey, 24 declined to complete the survey and provided a reason (nil commissioned n=14, service decommissioned n=4, insufficient resources to complete the survey n=3, declined for other reasons n=3) and 39 did not complete the survey and did not provide a reason. Of the remaining 49 LAs, it is possible that the DPH forwarded our email onto the relevant contact but did not copy us in or that some of these LAs simply did not commission a tier 2 weight management service for children.'

As some LAs are providers of obesity services, the 'Obesity Lead' may be involved in service provision. The relatively low response to the survey was identified as a limitation of the paper and we have added to the 'weaknesses' section that 'it is possible that in some LAs, details regarding the online survey did not reach the relevant person'. A FOI request was not submitted, and this has been

also been added as a limitation of the study. However, as these services are not mandatory, it is possible that some LAs did not respond as they did not commission a relevant service.

Comment 4: Were any of the negative responses (ie either no response, or a response stating no commissioned service) because their local service catered for a different age group to that being studied, ie not 4-16years, as some areas may have had services available for just older children or perhaps even younger children. Did respondents incorrectly assume that if they did not cater for this whole age range then they could not respond. Subsequent FOI surveys have identified that respondents often make mistakes in the interpretation of the questions, and without follow up telephone calls to clarify then mistakes are made.

Response: It was not within the remit of our study to look at preschool children or those over the age of 16 years.

The questions in the online survey tried to be as explicit as possible. Most LAs would not commission a service to cover the whole age range so we feel it is unlikely respondents assumed that if they did not cater for the whole age range, they couldn't respond. It is more likely that they didn't complete the study as they didn't commission a service or due to the time pressures on their work.

Comment 5: Also, do we know how many of these tier 2 services work in collaboration with a commissioned Tier 3 service (whether commissioned by the same LA or local CCG) and did this influence response rates or answers given? The fact that responses could be given in relation to a locally commissioned service even if it was outside of the time period being assessed in the study, may imply some responses were retrospective and related to services that were subsequently decommissioned. This might suggest a higher response rate than reflects activity, and actual figures are worse than presented. I am not sure whether it is a valid statistical method to determine the final sample size by saturation of information, ie when no new information seemed to emerge, giving a final sample size of 20.

Response: We do not have information on how many of the Tier 2 services work in collaboration with a commissioned Tier 3 service as this question did not fall within the remit of our study. In reality, there is almost no Tier 3 commissioning of services for childhood obesity in England although this may be addressed in the NHS Long Term Plan. The online survey data relates to a specified time period as we felt otherwise it would be difficult to conduct meaningful analyses of this quantitative data. In order to understand and capture the breadth of views on weight management programmes from local authorities with positive and negative experiences, we felt it would be too restrictive to limit participation to those that had a service within a specified time period as we also wanted to hear the views of those who may have decommissioned services. The main themes emerging from the interviews were similar after interviewing 20 participants which is why no further interviews were conducted. Achieving data saturation is a well-recognised method in qualitative analysis to indicate a satisfactory sample size has been achieved that provides accurate and generalisable data. Morse, J.M.: Data were saturated.... Qual. Health Res. 25(5), 587–588 (2015)

Comment 6: The results suggest that 64 LAs responded. Was this 64 x Dir of PH responded to the initial contact, or 64 responses from nominated obesity leads? If so, why did only 40 respond to the online survey? An FOI request would have gathered valuable information.

Response: The results section has been further clarified (see response to comment 3).

A FOI request was not submitted, and this is a limitation of the study. This has been added to the 'weaknesses' section 'A freedom of information (FOI) request was not submitted to obtain missing data and this is a limitation of the study.'

Comment 7: The fact that none of those completing the online survey knew of any evidence published in peer-reviewed journals demonstrating that their service was effective at improving BMI centile is very significant. However, it is worth noting that this is not the same as assuming that this

evidence is not there, as there is published data on the success of many child based T2 interventions, such as MEND, More Life, SHINE, etc.

Response: We do acknowledge that not knowing of evidence does not necessarily equate to evidence not existing. However, to the authors' best knowledge, the evidence of a clinically significant change in BMI z-score (of **>0.25**) with these programmes is limited and the proportion of participants completing these programme in the published literature is often relatively low.

MoreLife: Nobles et al report that '**47.1% of participants completed the MoreLife programme (mean attendance: 59.4 ± 26.7 %, mean BMI SDS change: -0.15 ± 0.22 units)**'.

- **Reference:** Nobles J, Griffiths C, Pringle A, Gately P. (2016). Design programmes to maximise participant engagement: a predictive study of programme and participant characteristics associated with engagement in paediatric weight management. *International Journal of Behavioural Nutrition and Physical Activity*, Volume 13, Number 1, Page 1

Shine: Shine appears to be a tier 3 weight management programme rather than a tier 2 weight management service which is the focus of our paper. Nobles et al reported that '*after 3 months, 95% of participants remained, with a mean BMI SDS reduction of .19 units (95% confidence interval: .17, .21)*' and acknowledge that '*The criteria for clinical significance are not agreed upon, but reductions of both .25 units and .50 units have been advocated and proposed as a goal for weight management. The findings of the current study indicate that only 35% of the cohort achieves a BMI SDS reduction .25 units and 5.6% achieve .50 unit reduction. Thus, achieving such thresholds is not likely to occur in the short term for the majority of WMP attendees, even at SHINE where the mean reductions in BMI SDS are greater than comparable programs.*' Although the long-term data from Shine appears promising, with BMI SDS scores >0.25 at 6, 9 and 12 months, the drop-out rate is large with only 53.9%, 41.2% and 30.8% participants retained at the respective time-points.

- Nobles, James & Radley, Duncan & Dimitri, Paul & Sharman, Kath. (2016). Psychosocial Interventions in the Treatment of Severe Adolescent Obesity: The SHINE Program. *Journal of Adolescent Health*. 59. 10.1016/j.jadohealth.2016.06.014.

Mend: Mend data is the closest to achieving a change in BMI z-score of >0.25. '*Participants in the intervention group had a reduced BMI z-score (-0.24; P < 0.0001) at 6 months when compared to the controls*'. '*At 12 months, children in the intervention group had reduced their BMI z-scores by 0.23 (P < 0.0001)*' and '*Mean attendance for the program was 86%*'.

- Sacher, Paul & Kolotourou, Maria & Chadwick, Paul & Cole, Tim & S Lawson, Margaret & Lucas, Alan & Singhal, Atul. (2010). Randomized Controlled Trial of the MEND Program: A Family-based Community Intervention for Childhood Obesity. *Obesity (Silver Spring, Md.)*. 18 Suppl 1. S62-8. 10.1038/oby.2009.433.

Comment 8: The cost data for each of the services is very interesting and valuable data, that I have not seen presented before. It would have been useful to see whether services that worked in collaboration with an established childhood tier 3 service (whether community based or secondary care) spent more or less than average on their tier 2 service

Response: Thank you for your comments on the cost data. Unfortunately, it was not within the remit of our study to look at costs where collaborations with tier 3 services were made. We imagine that if we had investigated this, it might be very difficult to disentangle costs associated with each service as resources (e.g. staff) might be shared between the tier 2 and tier 3 services. Some of the LAs included in our study were unable to give us cost data due to their budget covering other costs (e.g. one LA stated '*The service was part of a block contract covering a range of services, it is therefore not possible to pull out the cost*').

Comment 9: It is not clear from the text whether the 20 respondents to the telephone interviews were the named PH obesity lead or other. It does state that at least 2 were not, as they were service providers. I feel that the role of the respondent may well have given differing responses to similar questions, that was not based on differences in the services.

Response: The telephone interview participants were recruited from the online survey respondents. As some local authorities have 'in-house' contracts they can be both commissioners and service providers and so 'obesity leads' within the LA may be either of the latter. For clarity within our paper, we have amended the results section to state which interviews were conducted with service providers within the LA (see results section under 'Qualitative Data from Telephone Interviews' heading).

Comment 10: The fact that this highlights a consistent theme where commissioners have no standardised format for outcome data is significant and should be highlighted more in order to influence future commissioning. However, other responses need to be contextualised as we may be getting the opinions of non-experts in the field and we know that there are emotive and political agendas when it comes to childhood obesity.

Response: The future directions and conclusion has been amended to put greater emphasis on the need for standardisation of outcome data. The qualitative data does represent the views of the telephone interview participants and we have further contextualised these views within the current literature e.g. the work of Allender et al on the whole-systems approach has been included. Although the obesity leads in the LA are not 'experts' in the field of obesity, they are 'experts' in their experiences of these tier 2 weight management programmes in 'real life' and not in the trial setting. It is important to understand their views to help inform further research as they are arguable the most important link as will commission these services.

Comment 11: I think the section on future directions was good, with specific reference to the need for guidance on service specifications, contracts, collecting data, and analysing cost-benefit. However, more could have been made about the fact that childhood obesity services are not mandated, and as such the LGA have already stated that in times of financial restraint any non-mandated services, like obesity, will not be protected. There should have been discussion about the need for services to be mandated, along with a minimum level of service provision to complement the future directions suggested, together with the continuation of ring-fenced funding.

Response: Thank you for your comments on the future directions section.

There are only six mandated services which LAs must provide under public health:

- Weighing and measuring of children (NCMP)
- Health check assessment
- Conduct of health checks
- Sexual health services
- Public health advice service
- Protecting the health of the local population

Therefore, for this one service provision to be added as 'mandated' and to use this wording is very specific, and in our opinion wouldn't happen. As these services are commissioned by LAs rather than NHS, TAGs wouldn't apply. Recommendation 1 of NICE Public Health Guidance 47 (PH47) already recommends that 'family-based, multi-component lifestyle weight management services for children and young people are available' and that 'they should be provided as part of a locally agreed obesity care or weight management pathway.' NICE also advises that 'Directors of public health and public health teams working on obesity and child health and wellbeing' and 'Local authority commissioners' should act upon this recommendation.

So, as we don't feel these services would be made mandatory and NICE already recommends that LA commission lifestyle weight management programmes, we have instead added a sentence recognising the variation in service provision and the need to understand this and help support LAs in the commissioning and delivery of these services if cost-effectiveness is demonstrated;

'In the present format, tier 2 weight management services for overweight and obese children are very unlikely to have any impact on the prevalence of childhood obesity and peer-reviewed evidence of any long-term benefits even within the small numbers of children reached by these services, is weak. If these lifestyle weight management services are to be continued, clear thought needs to be given to the goals of the service and a more robust independent system needs to be developed to determine whether these goals are being met, whether the service is cost-effective and if it is the best use of limited resources in the current economic climate. Subsequently, if cost effectiveness is demonstrated, work needs to be undertaken to understand the variation in the provision of these services across England, such as through an 'Atlas of Variation,' and how LAs can be supported in the commissioning and delivery of these services, given that they are non-mandatory.

Comment 12: In the discussion, the authors make the observation that the reach of the services was only very small at 1.2%, however without giving population level context. ie data produced by More Life has previously demonstrated that for the large majority (62-98%) of obese adolescents, tracked from those children who were obese at a young age, ie relatively few healthy weight children made up the total of obese adolescents. So to save valuable resources (and by no means dismissing the importance of population-level public health campaigns for the masses) these young obese children need to be targeted, and so if the reach of 1.2% specifically targets a significant proportion of those obese children at a young age then it may have more long term impact than suggested. The lack of reach and lack of evidence for effectiveness is argued against the value of population level interventions such as the sugar taxes introduced in other countries. However, I feel that if the sugar tax is being proposed as an example of a public health intervention then a balanced argument should be given, exploring the limited benefits seen as a result of these taxes in countries that have introduced it and the potential disadvantages of this approach, otherwise again it is trying to influence the reader into preferring the public health approach. The WHO is also stated as having proposed other interventions for preventing childhood obesity, but the evidence base for these proposals are far from robust, without acknowledging this.

Response: The reach data was calculated using estimates of the prevalence of overweight or obese children within the relevant LA aged 4 to 16 years (this was estimated using NCMP data from Reception and Year 6 and National Statistics population data for children aged 4-16 years). The estimated mean reach of the service to overweight or obese children within a Local Authority was 1.2%. This indicates that the number of overweight or obese children which the service does not reach is 98.8% (based on estimates of prevalence of overweight or obese children in each LA) which implies that it is unlikely the service is targeting a significant number of children who are obese at a young age. If interventions can be designed which successfully engage a large proportion of young obese children and demonstrate long-term efficacy in improving weight status, then this would be a promising option, however, to date, this has not been achieved.

The future directions section and conclusion has been amended to reflect the limitations of the evidence regarding the sugar tax. It has also been amended to present a more balanced view with more emphasis on the need for the chosen obesity strategies to demonstrate cost-effectiveness. The conclusion has been changed from;

' In the future, consideration needs to be given as to whether evidence-based, population-level interventions should be given preferential funding rather than small-scale lifestyle weight management services with uncertainty regarding long-term effectiveness' to;

'In the future, obesity treatment or prevention programmes need to have robust systems in place to feedback programme outcomes and costs in a comparable transparent format to enable national independent oversight of the cost-effectiveness of different obesity strategies and direct future commissioning decisions'.

Reviewer 2: Miranda Pallan

Institution and Country: University of Birmingham

Comment: This is a very well written paper which contains information of interest to public health practitioners and healthcare commissioners in England.

Response: Thank you for your comments recognising the relevance and importance of this paper in the future commissioning and evaluation of childhood obesity services.

Comment : 1. I have a few amendments and suggestions to improve the paper further:

1. Introduction, end of first paragraph and second paragraph – provide some explanation of what CCGs, NHS England and Public Health England are for international readers

Response: The following paragraph has been added to the introduction section; ‘CCGs are responsible for the planning and commissioning of health care services for their local area and are assured by NHS England. In 2013, Public Health England was formed as a separate entity to NHS England as public health care transitioned from the NHS to Local Authorities under the Health and Social Care Act 2012.’

Comment: 2. Introduction, 4th paragraph – it would be helpful to clarify what you mean by evidence-base here (e.g. evidence underpinning the service provided in the local area)

Response: This has been clarified: ‘This mixed methods study uses quantitative methods (an online survey) to determine the evidence-base underpinning the local service provided, costs, reach, service usage and evaluation of tier 2 weight management programmes commissioned by Local Authorities across England for children aged 4-16 years.’

Comment: 3. Methods, online survey section – Please include more information on how this was developed – how did you determine the questions to include?

4. Methods, telephone interviews – Again, more detail on how the interview guides were developed, and what informed this process would be useful

Response: I have added a section as follows; ‘Design of Online Survey and Telephone Interview Guide: The online survey (Supplementary File 1) and interview guide (Supplementary File 2) were developed by RM, RJ, DS, JHS and RK. Development of the survey and interview guide were informed by the collective experiences of these clinicians and researchers in the field of childhood weight management and through addressing gaps in the current literature’.

Comment: 5. Results, Evidence-base for Tier 2 weight management service commissioned – it would be helpful to include examples of differences in outcome data collected by LAs

Response: I have amended this sentence as follows: ‘Due to heterogeneity in the way in which outcome data for change in weight status were reported by LAs (e.g. proportion who reduced or maintained their BMI z-score, number who ‘lost weight’, % of children who reduced their BMI z-score by 3%, only 6 or 12 month data), it was not possible to make any meaningful interpretations or comparisons of these data.’

Comment: 6. Results, Table 1 – the standard deviations given are very big and seem implausible (i.e. bigger than the mean in some cases); please check your figures for this

Response: The standard deviations are big due to large variations in the amounts LAs spent on their tier 2 service (from £5000 to £475000 per year). This is likely related to these services not being mandatory and so there is significant variation in what is provided for overweight or obese children in an LA.

Comment: 7. Results, Table 1 – in the foot note, please clarify that you are referring to the denominator used

Response: We are uncertain as to what footnote the reviewer is referring as Table 1 does not have a footnote?

Comment: 8. Qualitative data – for quotes given, please state if they came from a commissioner or provider.

Response: The qualitative data sources have now been clarified in the results section under the 'Qualitative Data from Telephone Interviews' section as follows; 'Twenty telephone interviews were conducted with LAs (18 commissioners, 2 service providers within the LA – Interview number 18 and 20)'.

Comment: 9. Results, outcome data - Give the numbers of LAs that collected data through performance management and those that collected data through evaluation. Also make it clear whether the data comes from the quantitative survey or the qualitative interviews.

Response: The results section is divided into two sections. The first section provides data from the quantitative survey and the second section provides data from the qualitative telephone interviews. In the online survey, data showed that service evaluations were conducted in 55% of Local Authorities and this is specified in the results section of the online survey under the heading 'Evidence base of Tier 2 Weight Management Service Commissioned'. We did not specifically ask whether the Local Authority used performance management data or service evaluation data to complete the online survey so cannot provide figures for this.

Comment: 10. Results, outcome data – the authors give examples of outcome measures that were unique to one or two LAs, from my reading it seems as if these probably relate to specific programme content. If this is the case, then it would be worth stating it in the manuscript

Response: We are unable to state whether these outcome measures related to specific programme content although this is possible. To avoid confusion, we have removed the following sentence: 'There were some outcome measures unique to one or two LAs e.g. **INT 5:** *'improved confidence with portion size'* or **INT 11:** *'the percentage of parents who have increased their confidence to read food labels from baseline'*'.

Comment: 11. Results, whole systems approach – this term can mean a whole variety of things, it would be worth clarifying what the Local Authority representatives interviewed meant by this

Response: Public Health England, the Local Government Association and the Association of Directors of Public Health are working with Leeds Beckett University to explore with LAs how to achieve a whole systems approach. All LAs will have been made aware of this work and presumably are referring to this when discussing the shift towards a whole systems approach to obesity. This work has been referenced in the results section.

Comment: 12. Discussion, 2nd paragraph – I would recommend separating the challenges associated with service provision and those associated with service evaluation.

Response: The sentence has been amended as suggested to separate the challenges associated with service provision and service evaluation;

'LAs identified many challenges facing their service in both service provision, through lack of engagement and lack of resources, and in service evaluation, through the questionable reliability of self-report data, lack of validated tools and difficulties in collecting long-term data'.

Comment: 13. Discussion, Meaning of the findings: implications for policy makers and clinicians, last paragraph – there is international literature on the use of whole systems approaches for obesity prevention, which could be incorporated here (see work done in Australia by Steve Allender and colleagues)

Response: Thank you for highlighting this paper. The discussion has been amended to include the work of Allender et al as follows:

'Allender et al describe a community's understanding of the complex causality of obesity through a causal loop diagram¹ and outline an obesity prevention trial aiming to use a whole systems community-led approach²'.

¹ Allender, Steven; Owen, Brynle; Kuhlberg, Jill; Lowe, Janette; Nagorcka-Smith, Phoebe; Whelan, Jill; et al. (2015): A Community Based Systems Diagram of Obesity Causes. PLOS ONE. 10(7): e0129683

² Allender, S., Millar, L., Hovmand, P., Bell, C., Moodie, M., Carter, R., Swinburn, B., Strugnell, C., Lowe, J., de la Haye, K., Orellana, L., ... Morgan, S. (2016). Whole of Systems Trial of Prevention Strategies for Childhood Obesity: WHO STOPS Childhood Obesity. *International journal of environmental research and public health*, 13(11), 1143. doi:10.3390/ijerph13111143

Comment: 14. Discussion, strengths and contextualisation, first paragraph – reference the evidence supporting NICE PH47 (links to the systematic reviews informing the guideline are available on the NICE website).

Response: The two reviews have now been cited.

Reviewer 3: Meliha Salahuddin

Institution and Country: University of Texas Health Science Centre at Tyler, Texas, USA

General

Comment 1: Please be consistent in the usage of "LA." The authors shift between the full and abbreviated form throughout the manuscript. Same for PHE.

Response: This has been amended in the revised manuscript.

Introduction

Comment 2: Please provide the full form for PH47 (page 3 line 32) and SMD (page 3 line 34).

Response: This has been amended to 'NICE public health guidance (PH47)' and 'standardised mean difference'.

Comment 3: Please clarify what is "ring-fenced public health budget" (page 3 line 46).

Public health in England moved from the NHS to Local Authorities in April 2013 and the LAs received an annual grant from the Department of Health which was deemed to be 'ring-fenced' and only to be used for public health functions. The ring-fenced budget has not been removed as planned in 2019. However, analysis by the King's Fund shows that "Spending on public health services by councils was 8 per cent lower in 2017/18 compared to 2013/14 (on a like-for-like basis)". This detail has been added instead. (<https://www.kingsfund.org.uk/projects/nhs-in-a-nutshell/spending-public-health>)

Methods

Comment 4: Please clarify whether any implementation science theories were used to evaluate the program such as RE-AIM framework (<http://www.re-aim.org/>), CFIR (<https://cfirguide.org/constructs/>), as examples. The authors may wish to contextualize their approach/analysis using these frameworks. The authors may find the following papers useful in determining the appropriate constructs along with the theory:

1. Proctor E, Silmere H, Raghavan R, et al. Outcomes for Implementation Research: Conceptual Distinctions, Measurement Challenges, and Research Agenda. *Administration and Policy in Mental Health and Mental Health Services Research*. 2011;38(2):65-76. doi:10.1007/s10488-010-0319-7
2. Rogers EM. *Diffusion of innovations*. 4th ed. New York: The Free Press; 1995.
3. Glasgow RE. *Translating Research to Practice*. Lessons learned, areas for improvement, and future directions. 2003;26(8):2451-6. doi:10.2337/diacare.26.8.2451
4. Glasgow RE, Vogt TM, Boles

SM. Evaluating the public health impact of health promotion interventions: the RE-AIM framework. *American Journal of Public Health*. 1999;89(9):1322-7. doi:10.2105/ajph.89.9.1322

Response: No specific implementation science theories were used in the design of this study so it would be difficult to retrospectively apply these. As we wanted to gain oversight of what was happening across England, we were keen to ensure that the online survey was not too laborious to complete and limited the number of questions to directly address our research aims. This meant that it was not possible to gain extensive details of the programme provided by each local authority.

Comment 5: Were any psychometric analyses done to evaluate the measures (e.g., reach)? Psychometric analysis would increase the methodological rigor of the study. Furthermore, could the measures be combined to form an index (Salahuddin, Barlow, Pont, Butte, & Hoelscher, 2018)?

Response: No psychometric analyses were done.

Comment 6: Sampling technique: The sampling technique for the semi-structured interview was not clear – it sounded arbitrary based on the comment “the final sample size was determined by the saturation of information when no new information seemed to emerge. This resulted in a final sample of 20 participants.” Was any power analysis conducted to determine the sample size? (page 6 lines 20-21). Why did the authors not conduct the semistructured interview with all 40 participants that completed the online survey?

Response: No power analyses were conducted as sample size was guided by data saturation (Ref: Guest, G., Bunce, A., Johnson, L.: How many interviews are enough? An experiment with data saturation and variability. *Field Methods* **18**(1), 59–82 (2006). The methods have been amended for clarity to state ‘The final sample size was determined when no new themes seemed to emerge from the interviews and data saturation was deemed to have been reached.’ Why 20 rather than 40 interviews: there is no advantage in conducting further interviews in qualitative studies when data saturation is reached. Also see Morse, J.M.: Data were saturated.... *Qual. Health Res.* 25(5), 587–588 (2015) referred to in replies to Reviewer one.

Comment 7: Here the authors are comparing 40 different LA commissioned tier 2 weight management services. It is not clear what the programs consisted of and whether they were all similar. Providing more information about the different program components is important. Otherwise, it is not clear whether the authors are comparing apples to oranges. Furthermore, the age range is broad (4-16 years). I would imagine weight management programs would be different for a 4-5-year-old versus a teenager. Additionally, it would be different depending on the severity of obesity (children with severe obesity (%BMI_{p95} > 120) versus children with obesity (%BMI ≥ p95 < 120).

Response: This study’s primary aim was to look at how commissioning decisions are made by weight management service providers and also to assess how they are evaluated by local teams. How different weight management interventions compare in terms of outcomes was outside of what we set out to assess. In-depth detail of each of the programmes components was not obtained so we are unable to provide this information. The only data collected was in response to the questions outlined in the online survey (See Supplementary File). As the programmes focused on in this study were ‘Tier 2’ weight management programmes, they are aimed at children who are overweight or obese rather than severely obese (Tier 3 services usually manage these children).

As it is not mandatory for LAs to commission a Tier 2 weight management service and evidence regarding long-term efficacy of these programmes is weak, the quantitative aspect of the study aimed to broadly understand what evidence was being used to justify commissioning decisions, how much LAs were spending on lifestyle weight management programmes for children (regardless of which age bracket they had chosen to target), how many children were engaging and completing interventions and whether LAs were evaluating their programmes. In order to fully evaluate the cost-effectiveness of the different programmes, in-depth details of the programmes and their data would be required and we did not obtain this information as we were aiming to achieve a broader oversight. This has been added as a limitation of the study in the ‘Strengths and Limitations’ section as follows;

'The change in weight status and cost data provided by LAs did not allow for meaningful statistical analyses to be conducted so it is not possible to comment on the cost-effectiveness of commissioned services'.

In a future study, it might be more feasible to conduct this at a regional level to allow for more intense investment in obtaining all the relevant data to provide a comprehensive analysis with meaningful statistical analyses.

Results

Comment 8: It is a critical limitation that the different weight-related outcomes measures were not evaluated as different programs measure different outcomes (page 7 lines 32-36).

Response: This limitation has been added to the 'Strengths and Limitations' box after the abstract as follows;

- 'The change in weight status and cost data provided by LAs did not allow for meaningful statistical analyses to be conducted so it is not possible to comment on the cost-effectiveness of commissioned services.'

However, as stated in response to comment seven, it was not our primary intention to compare outcomes such as efficacy and cost-effectiveness between weight management interventions but rather to understand commissioning processes.

Comment 9: It is not clear what is the difference between the first two rows in Table 1.

Response: Each LA has a different sized population. The first row states the cost to the LA. The second row contextualises this data according to the number of people living in the LA. The latter is perhaps more meaningful as enables comparisons in LA spending.

Discussion

Comment 10: The first paragraph of the Discussion typically includes a summary of the main findings.

Response: The first paragraph summarises the main findings on the quantitative research and the second paragraph summarises the main findings of the qualitative research.

Comment 11: It is well-documented in the literature that when it comes to obesity prevention, a holistic approach involving multiple sectors and multiple levels (individual, family, school, community) is effective. It is great that the local authorities identified this as a potential next step. Additionally, population-level intervention and measures would be more effective and particularly, intervening at an early age as obesity is known to track into adulthood.

Response: Yes, we agree that LAs identifying a whole systems approach bodes well for the future. We agree that finding a strategy to tackle or prevent obesity at an early age may provide a cost-effective option to reducing obesity prevalence in child and adulthood. We have added to the discussion the work of Allender et al on the whole systems approach to obesity.

VERSION 2 – REVIEW

REVIEWER	DR Matthew S Capehorn Rotherham Institute for Obesity (RIO) Clifton Medical Centre, Doncaster Gate Rotherham, S651DA
REVIEW RETURNED	24-Jun-2019

GENERAL COMMENTS	Despite my minor comments, most of which relate to the methodology and which have mostly been acknowledged and accepted in the discussion and limitations already, I think this is a really valuable paper and would add a great deal of value to the evidence base if published. Commissioners of child weight management services and service providers alike, have very little in the way of published data to reference such as cost, markers of success etc. The methodology initially involved an online survey, but was directed at those identified by local authority Directors of Public Health of PHE, or via suggestions from PHE, which I do not think is the most reliable, as appears to be the case. What was the response rate from the Directors of Public Health? Was it a case of some of them not responding, or was it a case of they did not even know who was responsible for commissioning child weight management services in their areas? This only identified 103/152 contacts. Of these only 40 completed the online survey. Why were Freedom Of Information requests not submitted (which is acknowledged) to increase data collection? Only 20 telephone interviews were conducted. I cannot see that it is clear as to how these were selected, given that only 17 of these had completed the online survey, and 2 were service providers. It would be interesting to know whether there were any objective differences in responses between service providers and commissioners. These semi-structured telephone interviews were carried out in Apr/Jun 2016, which was up to 2 years later than the earliest dates that the online survey related to (Apr 14 to Mar 15), during which time the services may well have changed, had changes in funding or service specification, or even been decommissioned. I do not feel that this has been sufficiently acknowledge. In the introduction there is a description of the 4-tiered model for weight management services. It states that Tier 3 and Tier 4 services are commissioned by a combination of clinical commissioning groups (CCGs) and NHS England, referencing the DH Working Party. Obvious most of this section relates to adult services, but this aside, the comment is only correct if dated, as prior to 2014 Tier 3 services were commissioned by a mix of CCGs and LA's and Tier 4 commissioned by NHSE. Following the working party, in 2014, CCGs became responsible for commissioning Tier 3 and it was always the intention that they would also commission Tier 4 as well "when ready". It was hoped this would be as early as 2016, but was actually rolled out in April 2017. Without clarification and/or dates then this statement might be misleading. However, with very little minor revision I think this could be a paper that has a lot of value.
---

REVIEWER	Meliha Salahuddin University of Texas Health Science Center at Tyler, USA
REVIEW RETURNED	07-Jun-2019

GENERAL COMMENTS	The authors have successfully addressed the comments. One issue still remains about whether any implementation theory was used to define the measures. If not, it should be addressed in limitation.
--

VERSION 2 – AUTHOR RESPONSE

Reviewer 1: Dr Matthew S Capehorn, Rotherham Institute for Obesity (RIO), UK

Comment 1: Despite my minor comments, most of which relate to the methodology and which have mostly been acknowledged and accepted in the discussion and limitations already, I think this is a really valuable paper and would add a great deal of value to the evidence base if published. Commissioners of child weight management services and service providers alike, have very little in the way of published data to reference such as cost, markers of success etc.

Response 1: Thank you for your comments recognising the relevance and importance of this paper in the future commissioning and evaluation of childhood obesity services.

Comment 2: The methodology initially involved an online survey, but was directed at those identified by local authority Directors of Public Health of PHE, or via suggestions from PHE, which I do not think is the most reliable, as appears to be the case. What was the response rate from the Directors of Public Health? Was it a case of some of them not responding, or was it a case of they did not even know who was responsible for commissioning child weight management services in their areas? This only identified 103/152 contacts. Of these only 40 completed the online survey. Why were Freedom Of Information requests not submitted (which is acknowledged) to increase data collection?

Response 2: As some Directors of Public Health forwarded the email onto the relevant person but did not cc me into this email, we are unable to state how many Directors of Public Health 'actioned' the email. We recognise the limitations of our recruitment method and have further acknowledged this in the limitations section by adding the underlined;

'Due to the method of recruitment to our study, it is possible that in some LAs, details regarding the survey did not reach the relevant person. A freedom of information (FOI) request was not submitted to obtain missing data and this is a limitation of the study.

We have already acknowledged that Freedom of information requests were not submitted in the limitations section of our paper.

Comment 3: Only 20 telephone interviews were conducted. I cannot see that it is clear as to how these were selected, given that only 17 of these had completed the online survey, and 2 were service providers. It would be interesting to know whether there were any objective differences in responses between service providers and commissioners.

Response 3: Recruitment to the telephone interviews was through the online survey and email contact with some of those who had declined to participate in the online survey. The latter group were included in order to capture the breadth of views on weight management programmes from local authorities with positive and negative experiences, including those who may have decommissioned services. Due to 'in-house' contracts, some Local Authorities were both commissioners and service providers. This has been acknowledged in the 'transparency statement' in the methods. After 20 telephone interviews, data saturation was deemed to have been reached and so no further interviews were conducted.

Comment 4: These semi-structured telephone interviews were carried out in Apr/Jun 2016, which was up to 2 years later than the earliest dates that the online survey related to (Apr 14 to Mar 15), during which time the services may well have changed, had changes in funding or service specification, or even been decommissioned. I do not feel that this has been sufficiently acknowledge.

Response 4: The LA's filled out the online survey between February and May 2016 based on services commissioned between April 2014 and March 2015. We requested Local Authorities to provide answers to the online survey, based on the service commissioned between April 2014 to March 2015, as this relates to a full financial year and would ensure that cost data was as accurate as possible enabling comparisons between LA's.

Whilst it is possible that the services may have changed, we do not feel that this impacts on the results of the survey itself nor on the data from the telephone interviews, as these data sets were reported on as separate entities in the results section. In addition, we have specifically stated in the methods that 'The interviews required participants to reflect on their experiences of tier 2 weight management services for school-aged children within their LA but was not confined to experiences within the time-period specified in the online survey of March 2014-April 2015. This enabled a broader representation of experiences from interview participants'. If there have been changes in the service since the time-period specified in the online survey, it will only enrich the responses participants provided in the telephone interviews.

Comment 5: In the introduction there is a description of the 4-tiered model for weight management services. It states that Tier 3 and Tier 4 services are commissioned by a combination of clinical commissioning groups (CCGs) and NHS England, referencing the DH Working Party. Obvious most of this section relates to adult services, but this aside, the comment is only correct if dated, as prior to 2014 Tier 3 services were commissioned by a mix of CCGs and LA's and Tier 4 commissioned by NHSE. Following the working party, in 2014, CCGs became responsible for commissioning Tier 3 and it was always the intention that they would also commission Tier 4 as well "when ready". It was hoped this would be as early as 2016, but was actually rolled out in April 2017. Without clarification and/or dates then this statement might be misleading.

Response 5: Thank you for these comments. We have amended the statement to include dates as follows: 'Clinical commissioning groups (CCG's) are responsible for commissioning Tier 3 services since 2014 and Tier 4 services since 2017'.

Comment 6: However, with very little minor revision I think this could be a paper that has a lot of value.

Response 6: Thank you.

Reviewer 3: Meliha Salahuddin, University of Texas, USA

Comment: The authors have successfully addressed the comments. One issue still remains about whether any implementation theory was used to define the measures. If not, it should be addressed in limitation.

Response: Thank you for your comments. The following sentence has been added to the weaknesses section: 'No implementation theories were used to evaluate programmes.'

VERSION 3 – REVIEW

REVIEWER	Dr Matthew S Capehorn Rotherham Institute For Obesity (RIO)
REVIEW RETURNED	11-Oct-2019

GENERAL COMMENTS	The study design still leaves the same limitations and weaknesses mentioned before, but this is adequately addressed in the
---

	discussion, and does not deter from additional and useful information for the commissioners and providers of lifestyle weight management programmes for children, so I think it does add to the evidence based and should be accepted.
--	--